# The utility of measures of anterior segment parameters of a Pentacam Scheimpflug tomographer in discriminating high myopic astigmatism from keratoconus

Ebenezer Zaabaar[1], Samuel Kyei [1]*, Maame Ama Amamoah Parkson Brew[2‡], Samuel Bert Boadi-Kusi[1‡], Frank Assiamah[3‡], Kofi Asiedu[4‡]

1 Department of Optometry and Vision Science, School of Allied Health Sciences, College of Health and Allied Sciences, University of Cape Coast, Cape Coast, Ghana, 2 Department of Imaging Technology and Sonography, School of Allied Health Sciences, College of Health and Allied Sciences, University of Cape Coast, Cape Coast, Ghana, 3 Eye Department, Korle-Bu Teaching Hospital, Accra, Ghana, 4 Eye Clinic Cosmopolitan Medical Center, Dworwulu, Accra, Ghana

☯ These authors contributed equally to this work.
‡ These authors also contributed equally to this work.
* skyei@ucc.edu.gh

**Data Availability Statement:** All relevant data are within the manuscript and its Supporting information files.

## Abstract

The study aimed to evaluate and compare anterior segment parameters between keratoconic eyes and eyes with high myopic astigmatism using Pentacam Scheimpflug tomography. This was a retrospective cross-sectional study that included sixty keratoconic eyes (thirty-two persons) and seventy-three eyes (forty-six persons) with high myopic astigmatism with mean ages 24.72 ± 11.65years and 26.60 ± 10.69years, respectively. Twenty-three parameters from the topographic map and fifteen parameters from the Belin-Ambrosió enhanced ectasia display map of the printouts of a Scheimpflug principle-based Pentacam tomographer were evaluated for their diagnostic accuracy using Receiver Operating Characteristic (ROC) curve. All parameters except cornea volume, anterior chamber volume, and anterior chamber angle indicated a significant difference between high myopic astigmatism and keratoconic eyes. The area under the receiver operating characteristic (AUROC) of eighteen Pentacam parameters was excellent (0.9–1.0) in discriminating keratoconus from high myopic astigmatism, out of which four {anterior minimum sagittal curvature (ant. Rmin), posterior minimum sagittal curvature (post. Rmin), maximum Ambrosió relational thickness (ART max) and total deviation value (D)} indicated excellent (>90%) sensitivity and specificity in addition to the excellent AUROC values. Topographic and Belin-Ambrosió enhanced ectasia display (BAD) maps of a Scheimpflug principle-based Pentacam tomographer are useful in enhancing the diagnosis of keratoconus and may also provide valuable information in effectively screening for keratoconus cases among refractive surgery candidates with high myopic astigmatism.

**Funding:** The authors received no specific funding for this work.

**Competing interests:** The authors have declared that no competing interests exist.

## Introduction

Keratoconus is a progressive, noninflammatory corneal thinning characterized by changes in the structure and organization of corneal collagen. It causes an uneven steepening of the central or paracentral zone of the cornea leading to irregular asymmetrical astigmatism, which cannot be fully corrected with glasses and therefore compromising the quality of life of the affected individuals [1–3]. The non-uniform steepening of the cornea that results from ectasia is the main cause of refractive error in eyes with keratoconus. However, oftentimes, increased axial length contributes to the myopic component of the refractive error [1–4]. Thus, it is not uncommon for keratoconus patients to develop high myopia with irregular astigmatism since most high myopias are axial [5].

Permanent correction of high myopia with refractive surgeries is increasing [6]. Laser-assisted in situ keratomileusis (LASIK) is the most common keratorefractive surgery [7]. This procedure reshapes the cornea by removing microscopic particles of tissue and may be associated with iatrogenic keratectasia (Post-LASIK ectasia) [8]. Keratoconic eyes are less resistant to deformation in post-LASIK ectasia due to focal weakening and significant reduction in corneal stiffness. This phenomenon results from variations in the expression of corneal epithelium and stroma-specific genes at the apex of the cone as well as changes in keratoconus-related proteome in non-cone regions of keratoconus corneas [9–11]. Accordingly, detecting keratoconus among refractive surgery candidates is crucial since removing tissue from an inherently weak cornea further weakens and threatens its integrity [12]. Besides, early diagnosis of keratoconus helps in prompt management of the condition to enhance the quality of life of patients.

Research has shown that high myopia irrespective of the degree of astigmatism can be considered as an alarming sign requiring further corneal examination to exclude any corneal abnormalities such as keratoconus [13]. A study on pediatric patients has also revealed cases of keratoconus that were misdiagnosed as meridional amblyopia secondary to myopic astigmatism due to the unavailability of corneal tomography measurements [14]. It is therefore important to always screen high myopic patients, particularly those considering refractive surgery, to identify high-risk corneas.

Despite these familiar clinical situations, the relationship between keratoconus and high myopia has not been extensively studied and investigated in the ophthalmic literature. There is a lag in the clinical uptake of corneal tomographic measurements for eyes with high myopic astigmatism. Besides, the majority of studies that attempted to explore the dynamics of anterior segment parameters in keratoconus were carried out among Asians and Caucasians [15–19], and to the best of our knowledge, this is the only study that has compared anterior segment parameters between eyes with keratoconus and eyes with high myopic astigmatism among black Africans.

The criteria for the diagnosis and classification of keratoconus are based on anterior corneal curvature data derived with the Placido-based corneal topography [15, 16]. However, studies have suggested that early changes in eyes with keratoconus are also present on the posterior corneal surface [15–17]. Scheimpflug imaging measures the entire cornea thickness by determining the front and back surfaces of the cornea with a rotating Scheimpflug camera. Several studies have used Scheimpflug tomography to compare anterior segment parameters between normal eyes and eyes with keratoconus [15, 18, 19], but the use of Scheimpflug imaging to determine a difference between keratoconus and high myopic astigmatism is inadequately investigated. This study aimed at comparing Pentacam Scheimpflug corneal tomography findings between keratoconus eyes and high myopic astigmatic eyes and to determine the sensitivity and specificity of these parameters in discriminating keratoconus from high myopic astigmatism.

## Materials and methods

The study employed a retrospective cross-sectional design and followed the tenets of the Declaration of Helsinki. The investigation covered a 7-year period from January 2014 to December 2020. Ethical approval for the current study protocol was obtained from the Institutional Review Board (IRB) of the University of Cape Coast, Ghana (UCCIRB/CHAS/2018/65). Since medical records were reviewed retrospectively and identifying particulars of patients' details concealed, patients' consents were not needed and were therefore waived by the IRB. All data and records generated throughout the study were handled with strict confidentiality in alignment with the University of Cape Coast institutional policies.

Clinical diagnosis of keratoconus was made if eyes had irregular corneas determined by distorted keratometric mires, distorted red reflex on retinoscopy or ophthalmoscopy or both, and at least one of the following slit-lamp biomicroscopic findings: Vogt striae, Fleischer ring of more than 2.0mm arc length, and corneal scarring consistent with keratoconus [20]. All diagnoses were made by an ophthalmologist with expertise in cornea and external eye disease. Records of patients diagnosed with ocular conditions other than keratoconus were excluded. Eyes were assigned to the control group of high myopic astigmatism if they had no history of ocular surgery, eye pathology, or irregular cornea patterns. Records of all eyes with a history of contact lens wear or any corneal intervention before Pentacam scans were excluded.

Records of patients examined with the Scheimpflug principle-based Pentacam corneal tomographer (Wavelight—Allegro Oculyzer, GmbH, Erlangen, Germany) were reviewed for topographic parameters and parameters of the BAD maps. Parameters studied included keratometry readings, topographic astigmatism, corneal eccentricity in the central 6mm, average radius of curvature between the 6mm and 9mm zone center (Rper), and minimum sagittal curvature (Rmin) for anterior and posterior cornea surfaces. Additionally, pachymetry, cornea volume, anterior chamber volume, anterior chamber angle, anterior chamber depth, keratometric power deviation (KPD), and Belin-Ambrosió enhanced ectasia display (BAD) readings were recorded.

Corneal thickness measurements were taken for multiple points (apex, thinnest location over anterior cornea surface, and pupil center). Cornea volume was reported for a diameter of 10mm centered on the anterior corneal apex. Anterior chamber depth was measured as the distance from the corneal endothelium to the anterior surface of the lens capsule. The anterior chamber volume was computed from the endothelium down to the iris and lens over a 12mm diameter centered on the anterior corneal apex. The anterior chamber angle recorded was the smallest in the horizontal position calculated from the Scheimpflug image. The float option of the best fit sphere served as a reference surface for front and back elevation data measurements, and the diameter of the reference surface was 8 mm. The front elevation was determined as the maximum difference in anterior corneal elevation between the best-fit sphere (BSF) and the enhanced best fit sphere obtained with the BAD display software. Back elevation was also determined as the maximum differential change in posterior corneal elevation between the best fit sphere (BFS) and the enhanced BFS obtained with the BAD display software. The progression index was computed as the average progression value at the different pachymetric rings.

Visual acuity was measured using a LogMAR chart. Spherical refractive error and total astigmatism were determined objectively using an auto refractometer and subjectively by the maximum plus to maximum visual acuity method at 6m. Patients who had sphero-cylindrical refractive errors with spherical components greater than -6.00D were considered high myopic astigmats.

Data were analyzed using the Statistical Package for Social Sciences (SPSS) for Windows, version 22.0 (Armonk, NY: IBM Corp.), and a p-value of less than 0.05 was considered statistically significant. Independent samples t-test was used to find the mean difference in parameters between the two groups. ROC curve was used to plot sensitivity (true positive rate) against 1-specificity (false positive rate) for the different thresholds of the diagnostic test, and the overall diagnostic accuracy of the test evaluated with the area under the ROC curve (AUROC). The AUROC curve ranges from 1 (100%) to 0.5 (50%). An AUROC curve of 1 (100%) indicates perfect discrimination, and an AUROC curve of 0.5 (50%) denotes a test that is only ever accurate by chance—a completely bad classification. Within this range, 0.9–1.0, 0.8–0.9, 0.7–0.8, 0.6–0.7 and 0.5–0.6 indicate excellent, good, fair, poor and very poor discrimination respectively [21].

## Results and discussion

Sixty keratoconic eyes of thirty-two persons with a mean uncorrected visual acuity (UCVA) of 0.94 ± 0.45 were involved in the study. There were ten (31.25%) females and twenty-two (68.75%) males with a mean age of 24.72 ± 11.65 years (Range = 7–69 years).

The high myopic astigmatic group comprised 73 eyes of forty-six persons with a mean UCVA of 1.19 ± 0.40. The group consisted of 21(45.65%) females and 25(54.35%) males with a mean age of 26.60 ± 10.69 years (Range = 10–51 years).

There was no significant difference between the two groups regarding gender ($\chi2$ = 2.39, p = 0.12) and age, t (121.29) = -0.96, p = 0.34. However, UCVA differed substantially between the two groups t (120.17) = -3.42, p = 0.001. The means and ranges of all Pentacam parameters obtained from the topographic and Belin-Ambrosió enhanced ectasia display maps are shown in Tables 1 and 2. All parameters except cornea volume, anterior chamber volume, and anterior chamber angle indicated a significant difference between keratoconic and myopic astigmatic eyes. Table 3 presents the mean refractive errors of both groups.

Results of the Receiver Operating Characteristic (ROC) curve analysis are shown in Tables 4 and 5.

AUROC values of eighteen Pentacam parameters were excellent (0.9–1.0) in discriminating keratoconus from high myopic astigmatism, out of which four (Front Rmin, Back Rmin, ARTmax, and D) indicated excellent (>90%) sensitivity and specificity in addition.

There was a significant difference between keratoconic eyes and eyes with high myopic astigmatism concerning all parameters except corneal volume, anterior chamber volume, and anterior chamber angle. Of all the parameters evaluated on the topographic map, steepest front keratometry reading (Ksteep-front), mean front keratometry reading (Kmean-front), steepest back keratometry reading (Ksteep-back), maximum keratometry reading (Kmax), astigmatism (back), thinnest corneal thickness (TCT), minimum front sagittal curvature (Rmin-front) and minimum back sagittal curvature (Rmin-back) indicated excellent predictive accuracy.

In the present study, a cutoff value of 485µ for TCT yielded 83% sensitivity and 82% specificity in distinguishing between keratoconus and high myopic astigmatism, which further implicates corneal stromal thinning as the hallmark of keratoconus. Earlier studies reported cutoff points of TCT ranging from 489 µ to 506 µ [15, 20, 22, 23]. The outstanding diagnostic efficacy of TCT corroborates the results of similar studies that compared keratoconus eyes with normal emmetropic eyes [15, 24, 25].

The minimum sagittal curvature of the front and back corneal surfaces showed excellent predictive ability. A cutoff value of 7.03mm for the minimum front sagittal curvature had 90% sensitivity and 90% specificity in discriminating keratoconus from high myopic astigmatism.

**Table 1. Comparison of mean parameters of topographic maps between keratoconus and high myopic astigmatic eyes.**

| Pentacam parameter | Keratoconus Mean ± SD (Range) | High Myopic astigmatism Mean ± SD (Range) | P |
|---|---|---|---|
| Kflat (Front) | 50.47 ± 7.9 (39.50–70) | 42.82 ± 2.68 (38.7–57.8) | <0.001 |
| Ksteep (Front) | 57.09 ± 9.1 (42.7–76) | 44.65 ± 2.97 (41.1–60.2) | <0.001 |
| Kmean (Front) | 53.31 ± 8.11 (42–72.9) | 43.7 ± 2.73 (39.9–59) | <0.001 |
| Kmax | 62.54 ± 12.04 (44.6–89.5) | 45.39 ± 4.21 (41.4–69.5) | <0.001 |
| Astigmatism (Front) | -6.01 ± 4.17 {-0.6-(-25.7)} | -1.84 ± 1.43 {-0.2-(-7.1)} | <0.001 |
| Eccentricity (Front) | -0.92 ± 0.49 {-0.07- (-2.04)} | -0.34 ± 0.22 {-0.01-(-0.96)} | <0.001 |
| Rper (Front) | 7.69 ± 0.55 (6.33–8.8) | 8.15 ± 0.28 (7.46–8.82) | <0.001 |
| Kflat (Back) | -7.18 ± 1.74 {-2.5-(-11.2)} | -6.03 ± 0.47 {-5.4-(-8.8)} | <0.001 |
| Ksteep (Back) | -8.6 ± 1.9 {-5.9-(-14.4)} | -6.4 ± 0.56 {-5.8-(-9.4)} | <0.001 |
| Kmean (Back) | -7.79 ± 1.74 {-4.0-(-11.9)} | -6.21 ± 0.5 {-5.6-(-9.1)} | <0.001 |
| Astigmatism (Back) | -1.45 ± 1.3 {-0.3-(-6.7)} | -0.45 ± 0.73 {0.0- (-6.3)} | <0.001 |
| Eccentricity (Back) | -0.97 ± 0.46 {-0.03- (-1.72)} | -0.56 ± 0.13 {-0.23- (-1.11)} | <0.001 |
| Rper (Back) | 6.47 ± 0.61 (4.72–8.81) | 6.78 ± 0.27 (6.04–7.27) | 0.001 |
| Pupil center | 455.45 ± 53.97 (246–536) | 518.64 ± 35.36 (412–591) | <0.001 |
| Pachy apex | 446.98 ± 57.49 (267–556) | 517.59 ± 36.91 (407–590) | <0.001 |
| TCT | 407.25 ± 81.67 (109–530) | 513.52 ± 37.14 (403–589) | <0.001 |
| Cornea vol | 58.09 ± 5.06 (46.7–69.1) | 57.43 ± 3.89 (49.5–65.5) | 0.41 |
| AC vol | 184.35 ± 47.95 (68.4–349) | 180.78 ± 36.51 (66–260) | 0.64 |
| ACD | 4.11 ± 0.41 (3.26–4.89) | 3.68 ± 0.33 (3.07–4.26) | <0.001 |
| ACA | 41.31 ± 11.99 (16.8–88.4) | 40.75 ± 8.51 (25.9–82.9) | 0.76 |
| KPD | 2.37 ± 1.29 (0.3–6.0) | 1.13 ± 0.35 (0.7–3.4) | <0.001 |
| Rmin (Front) | 5.5 ± 1.32 (0.43–8.47) | 7.49 ± 0.53 (4.86–8.15) | <0.001 |
| Rmin (Back) | 4.05 ± 1.09 (2.05–6.29) | 6.1 ± 0.56 (3.33–6.75) | <0.001 |

K: keratometry reading, Rper: average radius of curvature between the 6mm and 9mm zone center, pachy apex: corneal thickness at the apex, TCT: thinnest corneal thickness; cornea vol.: corneal volume, AC vol.: anterior chamber volume, ACD: anterior chamber depth, ACA: anterior chamber angle, KPD: keratometric power deviation, Rmin: minimum sagittal curvature.

Also, a cutoff of 5.64mm for the minimum back sagittal curvature had 92% sensitivity and 90% specificity in differentiating keratoconus from high myopia. These results are consistent with those reported by Orucoglu and Toker, who also found minimum front and back sagittal curvatures as excellent in differentiating keratoconus from normal eyes [15].

Eccentricity coefficient is an index that describes how the corneal curvature changes from the central region to the peripheral region [26]. Consistent with an earlier investigation [15], front and back eccentricity coefficients did not indicate excellent predictive accuracy in differentiating keratoconus from high myopic astigmatism. A cornea with astigmatism has both oblate and prolate meridians [27]. The prolate axis has a more negative eccentricity value and the oblate axis, a less negative or positive eccentricity value. When the oblate meridian dominates, the mean eccentricity value becomes less negative or positive, and when the prolate meridian dominates, the mean eccentricity value becomes more negative [27]. Against this backdrop, the high total astigmatism values recorded in the two groups of the current study may not provide reliable information about corneal eccentricity. Therefore, the eccentricity coefficient is not specific for discriminating keratoconus from high myopic astigmatism and must be considered with the apex position of the cone and the magnitude of astigmatism.

Out of the fifteen parameters evaluated on the Belin-Ambrosió enhanced ectasia display map, fourteen demonstrated excellent discrimination. Consistent with the current study, Orucoglu and Toker [15] also reported fourteen BAD display map parameters as being

**Table 2. Comparison of mean parameters of Belin-Ambrosió enhanced ectasia display (BAD) maps between keratoconus and myopic astigmatic eyes.**

| Pentacam parameter | Keratoconus Mean ± SD (Range) | High Myopic astigmatism Mean ± SD (Range) | P |
|---|---|---|---|
| Front difference | 21.68 ± 16.12 (1.0–68.0) | 4.14 ± 2.29 (1.0–16.0) | <0.001 |
| Back difference | 45.75 ± 32.1 (2.0–140.0) | 5.20 ± 4.29 (0.0–27) | <0.001 |
| Dist.Apex.Th | 1.02 ± 0.66 (0.22–4.12) | 0.73 ± 0.29 (0.06–1.24) | 0.002 |
| Front elevation | 49.57 ± 30.06 (8.0–143.0) | 12.23 ± 9.57 (2.0–54.0) | <0.001 |
| Back elevation | 102.7 ± 66.12 (12.0–306.0) | 21.52 ± 14.95 (6.0–100.0) | <0.001 |
| ProgMin | 2.69 ± 2.62 (0.56–13.71) | 0.74 ± 0.37 (0.40–3.16) | <0.001 |
| ProgMax | 6.46 ± 7.26 (1.29–30.19) | 1.28 ± 0.51 (0.81–4.69) | <0.001 |
| ProgAvg | 2.9 ± 2.73 (0.00–15.27) | 0.94 ± 0.27 (0.39–2.35) | <0.001 |
| ARTmax | 121.41 ± 82.95 (0.00–375) | 430.43 ± 107.11 (86.0–724.0) | <0.001 |
| Df | 14.11 ± 11.13 (-0.55–44.34) | 1.59 ± 3.76 (-1.3–25.76) | <0.001 |
| Db | 18.96 ± 36.25 (-0.91–265.52) | 0.31 ± 2.81 (-1.4–19.79) | <0.001 |
| Dp | 15.15 ± 19.4 (-6.12–97.16) | 0.52 ± 2.31 (-1.61–15.28) | <0.001 |
| Dt | 6.29 ± 8.61 (0.24–60.94) | 0.8 ± 1.28 (-1.34–5.16) | <0.001 |
| Da | 3.04 ± 1.09 (0.17–4.46) | 0.53 ± 1.42 (-7.0–3.67) | <0.001 |
| D | 14.64 ±15.41 (1.05–112.43) | 1.46 ± 2.35 (-0.64–15.95) | <0.001 |

Dist.Apex.Th: distance from corneal apex to thinnest location, ProgMin/Max/Avg: progression index, ARTmax: maximum Ambrosió relational thickness, Df: deviation of front elevation difference map, Db: deviation of back elevation difference map, Dp: deviation of average pachymetric progression, Dt: deviation of minimum thickness, Da: deviation of ARTmax, D: total deviation value.

outstanding in diagnostic ability. Fam and Lim [28] reported the clinical relevance of front and back elevation parameters for the detection of keratoconus and suspected or subclinical keratoconus eyes. Earlier investigations have indicated excellent diagnostic efficacies of front and back elevation measures in diagnosing keratoconus [15, 17, 29]. The results of these previous studies are consistent with the outcome of the current investigation. However, the cutoff values for detecting eyes with keratoconus in this study were higher compared to those of earlier studies. This could be due to variability in the control group as well as variability in the stages of keratoconus cases investigated.

The pachymetric progression index calculates the change in corneal thickness over 360 degrees of the cornea. The progression value at each meridian from the thinnest point is defined as progression index and the average of all meridians is illustrated by Prog-Avg [30–32]. Doctor and colleagues [33] reported the clinical significance of a rapid rate of pachymetric progression in distinguishing keratoconus from normal eyes. Other studies [20, 22, 30] have also reported excellent predictive accuracy in using pachymetric progression indices to discriminate keratoconus from normal eyes. In the current study, maximum pachymetric progression (Prog-Max) provided the best combination of sensitivity (93%) and specificity (82%) in predicting keratoconus. This corroborates outcomes reported in previous reviews [20, 22, 30]. Cutoffs for the progression indices also compare favorably to results of prior studies.

Ambrosió relational thickness (ART) is the ratio between the thinnest point and progression index. It includes ART max, ART min, and ART avg [33]. Several studies [15, 18, 22, 24,

**Table 3. Mean refractive errors of keratoconic and high myopic astigmatic eyes.**

| Component of error | Keratoconus Mean ± SD (Range) | High Myopic Astigmatism Mean ± SD (Range) | P |
|---|---|---|---|
| Sphere (myopic) | -7.82 ± 5.94 {-0.25 –(-)30.0} | -11.4 ± 3.37 {-6.25 –(-)18.5} | <0.001 |
| Cylinder | -5.69 ± 2.84 {-0.25 –(-)14} | -2.17 ± 1.63 {-0.25 –(-)8.25} | <0.001 |

**Table 4. Receiver Operating Characteristic (ROC) curve analysis of topographic parameters of keratoconic and high myopic astigmatic eyes.**

| Parameters | AUC | SE | 95%CI | P | Cutoff | Sensitivity | Specificity |
|---|---|---|---|---|---|---|---|
| Kflat (Front) | 0.818 | 0.038 | 0.743–0.893 | <0.001 | 45.10 | 0.65 | 0.92 |
| Ksteep (Front) | 0.927 | 0.024 | 0.880–0.975 | <0.001 | 46.10 | 0.92 | 0.89 |
| Kmean (Front) | 0.900 | 0.028 | 0.846–0.954 | <0.001 | 45.25 | 0.83 | 0.89 |
| Kmax | 0.953 | 0.018 | 0.919–0.988 | <0.001 | 46.40 | 0.97 | 0.86 |
| Astigmatism (Front) | 0.861 | 0.033 | 0.796–0.925 | <0.001 | 2.65 | 0.80 | 0.86 |
| Eccentricity (Front) | 0.827 | 0.037 | 0.754–0.900 | <0.001 | -0.39 | 0.77 | 0.60 |
| Rper (Front) | 0.761 | 0.041 | 0.680–0.842 | <0.001 | 8.00 | 0.65 | 0.64 |
| Kflat (Back) | 0.760 | 0.048 | 0.666–0.854 | <0.001 | -6.05 | 0.77 | 0.56 |
| Ksteep (Back) | 0.901 | 0.030 | 0.843–0.960 | <0.001 | -6.75 | 0.87 | 0.89 |
| Kmean (Back) | 0.824 | 0.042 | 0.742–0.906 | <0.001 | -6.45 | 0.75 | 0.84 |
| Astigmatism (Back) | 0.911 | 0.026 | 0.861–0.961 | <0.001 | 0.55 | 0.85 | 0.85 |
| Eccentricity (Back) | 0.774 | 0.049 | 0.678–870 | <0.001 | -0.59 | 0.77 | 0.64 |
| Rper (Back) | 0.690 | 0.047 | 0.598–0.783 | <0.001 | 6.79 | 0.68 | 0.55 |
| Pupil center | 0.867 | 0.031 | 0.807–0.927 | <0.001 | 505.50 | 0.80 | 0.75 |
| Pachy apex | 0.859 | 0.033 | 0.795–0.924 | <0.001 | 497.00 | 0.80 | 0.75 |
| TCT | 0.919 | 0.023 | 0.874–0.963 | <0.001 | 485.00 | 0.83 | 0.82 |
| Cornea vol. | 0.530 | 0.051 | 0.429–0.630 | 0.555 | 57.45 | 0.52 | 0.49 |
| AC vol. | 0.522 | 0.051 | 0.422–0.623 | 0.656 | 180 | 0.55 | 0.52 |
| ACD | 0.785 | 0.039 | 0.709–0.862 | <0.001 | 3.99 | 0.63 | 0.81 |
| ACA | 0.517 | 0.051 | 0.416–0.618 | 0.740 | 40.25 | 0.57 | 0.51 |
| KPD | 0.833 | 0.041 | 0.752–0.914 | <0.001 | 1.25 | 0.78 | 0.86 |
| Rmin (Front) | 0.939 | 0.023 | 0.894–0.985 | <0.001 | 7.03 | 0.90 | 0.90 |
| Rmin (Back) | 0.950 | 0.019 | 0.913–0.986 | <0.001 | 5.64 | 0.92 | 0.90 |

K: keratometry reading, Rper: average radius of curvature between the 6mm and 9mm zone center, pachy apex: corneal thickness at the apex, TCT: thinnest corneal thickness; cornea vol.: corneal volume, AC vol.: anterior chamber volume, ACD: anterior chamber depth, ACA: anterior chamber angle, KPD: keratometric power deviation, Rmin: minimum sagittal curvature.

25, 30] have reported ART max as a valid diagnostic index in discriminating keratoconic eyes from normal eyes. In the current study, the ART max produced the leading blend of sensitivity (93%) and specificity (90%) in discriminating keratoconus from high myopic astigmatism. The cutoff value of 300.50 was similar to those reported by earlier investigations [15, 22, 25].

The D parameters denote the standard deviation from the mean of the normative database. They are changes in anterior elevation from standard to enhanced reference surface, changes in posterior elevation, corneal thickness at the thinnest point, thinnest point displacement, and pachymetric progression {Df(front), Db (back), Dp (pachymetry progression), Dt (thinnest value), and Da (thinnest displacement)}. The total deviation (D) is computed by considering all 5 parameters and running a linear regression analysis against a standard database of normal and keratoconus corneas [33]. In common with what is reported in a previous study, it was found that all D parameters showed excellent precision in the diagnosis of keratoconus [11].

A limitation of the study is that eyes with early-stage keratoconus were not separated from eyes with later-stage keratoconus. Besides, the study is retrospective and non-randomized therefore further longitudinal and randomized studies are needed to affirm the finding of the study. Of all the Pentacam parameters evaluated, ARTmax, D, front Rmin, and back Rmin were the most sensitive and specific in discriminating keratoconus from high myopic astigmatism since they had excellent sensitivity and specificity in addition to excellent AUROC.

**Table 5. Receiver Operating Characteristic (ROC) curve analysis of parameters on Belin-Ambrosió enhanced ectasia display (BAD) maps of keratoconic and high myopic astigmatic eyes.**

| Parameters | AUC | SE | 95%CI | P | Cutoff | Sensitivity | Specificity |
|---|---|---|---|---|---|---|---|
| Front difference | 0.900 | 0.033 | 0.824–0.953 | <0.001 | 5.50 | 0.83 | 0.86 |
| Back difference | 0.926 | 0.025 | 0.877–0.975 | <0.001 | 8.50 | 0.87 | 0.85 |
| Dist.Apex.Th | 0.632 | 0.049 | 0.536–0.729 | 0.009 | 0.73 | 0.63 | 0.51 |
| Front elevation | 0.932 | 0.021 | 0.891–0.973 | <0.001 | 17.5 | 0.92 | 0.86 |
| Back elevation | 0.952 | 0.019 | 0.915–0.988 | <0.001 | 26.5 | 0.95 | 0.81 |
| ProgMin | 0.926 | 0.025 | 0.878–0.974 | <0.001 | 0.88 | 0.88 | 0.85 |
| ProgMax | 0.963 | 0.014 | 0.935–0.991 | <0.001 | 1.49 | 0.93 | 0.82 |
| ProgAvg | 0.900 | 0.038 | 0.803–0.950 | <0.001 | 1.16 | 0.85 | 0.88 |
| ARTmax | 0.979 | 0.011 | 0.957–1.000 | <0.001 | 300.50 | 0.93 | 0.90 |
| Df | 0.900 | 0.031 | 0.828–0.950 | <0.001 | 2.46 | 0.83 | 0.88 |
| Db | 0.910 | 0.028 | 0.856–0.964 | <0.001 | 2.01 | 0.83 | 0.93 |
| Dp | 0.891 | 0.035 | 0.821–0.960 | <0.001 | 1.92 | 0.85 | 0.89 |
| Dt | 0.919 | 0.023 | 0.875–0.963 | <0.001 | 1.65 | 0.83 | 0.81 |
| Da | 0.928 | 0.022 | 0.885–0.972 | <0.001 | 1.85 | 0.85 | 0.88 |
| D | 0.957 | 0.017 | 0.923–0.991 | <0.001 | 2.91 | 0.90 | 0.90 |

Dist.Apex.Th: distance from corneal apex to thinnest location, ProgMin/Max/Avg: progression index, ARTmax: maximum Ambrosió relational thickness, Df: deviation of front elevation difference map, Db: deviation of back elevation difference map, Dp: deviation of average pachymetric progression, Dt: deviation of minimum thickness, Da: deviation of ARTmax, D: total deviation value.

Topographic and Belin-Ambrosió enhanced ectasia display maps of a Scheimpflug principle-based Pentacam corneal topographer can be useful in the diagnosis of keratoconus; they may also provide valuable information in screening for keratoconus cases among refractive surgery candidates with high myopic astigmatism.

## Supporting information

**S1 Dataset.**
(SAV)

## Acknowledgments

The authors are grateful to the research assistants for their immense contribution to the collection of the data.

## Author Contributions

**Conceptualization:** Ebenezer Zaabaar, Samuel Kyei.

**Data curation:** Ebenezer Zaabaar, Samuel Kyei, Maame Ama Amamoah Parkson Brew, Samuel Bert Boadi-Kusi, Frank Assiamah, Kofi Asiedu.

**Formal analysis:** Ebenezer Zaabaar, Samuel Kyei, Maame Ama Amamoah Parkson Brew, Samuel Bert Boadi-Kusi, Frank Assiamah, Kofi Asiedu.

**Funding acquisition:** Ebenezer Zaabaar, Samuel Kyei, Maame Ama Amamoah Parkson Brew, Samuel Bert Boadi-Kusi, Frank Assiamah, Kofi Asiedu.

**Investigation:** Ebenezer Zaabaar, Samuel Kyei, Maame Ama Amamoah Parkson Brew, Samuel Bert Boadi-Kusi, Frank Assiamah, Kofi Asiedu.

**Methodology:** Ebenezer Zaabaar, Samuel Kyei, Maame Ama Amamoah Parkson Brew, Samuel Bert Boadi-Kusi, Frank Assiamah, Kofi Asiedu.

**Project administration:** Ebenezer Zaabaar, Samuel Kyei, Maame Ama Amamoah Parkson Brew, Samuel Bert Boadi-Kusi, Frank Assiamah, Kofi Asiedu.

**Resources:** Ebenezer Zaabaar, Samuel Kyei, Maame Ama Amamoah Parkson Brew, Samuel Bert Boadi-Kusi, Frank Assiamah, Kofi Asiedu.

**Software:** Ebenezer Zaabaar, Samuel Kyei, Maame Ama Amamoah Parkson Brew, Samuel Bert Boadi-Kusi, Frank Assiamah, Kofi Asiedu.

**Supervision:** Ebenezer Zaabaar, Samuel Kyei.

**Validation:** Ebenezer Zaabaar, Samuel Kyei.

**Visualization:** Ebenezer Zaabaar, Samuel Kyei.

**Writing – original draft:** Ebenezer Zaabaar, Samuel Kyei, Maame Ama Amamoah Parkson Brew, Samuel Bert Boadi-Kusi, Frank Assiamah, Kofi Asiedu.

**Writing – review & editing:** Ebenezer Zaabaar, Samuel Kyei, Maame Ama Amamoah Parkson Brew, Samuel Bert Boadi-Kusi, Frank Assiamah, Kofi Asiedu.

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
