## [Decision Letter · Decision Letter 0]

3 Aug 2021

PONE-D-21-20839

The utility of measures of anterior segment parameters of pentacam scheimpflug tomographer in discriminating high myopic astigmatism from keratoconus

PLOS ONE

Dear Dr. Kyei,

Thank you for submitting your manuscript to PLOS ONE. After careful consideration, we feel that it has merit but does not fully meet PLOS ONE’s publication criteria as it currently stands. Therefore, we invite you to submit a revised version of the manuscript that addresses the points raised during the review process.

We look forward to receiving your revised manuscript.

Kind regards,

Michael Mimouni

Academic Editor

PLOS ONE

Journal Requirements:

3. In the Methods section of the manuscript please additional information regarding how the participants were recruited for the study included any eligibility criteria applied.

4. You indicated that you had ethical approval for your study. In your Methods section, please ensure you have also stated whether you obtained consent from parents or guardians of the minors included in the study or whether the research ethics committee or IRB specifically waived the need for their consent.

Reviewers' comments:

Reviewer's Responses to Questions

**Comments to the Author**

1. Is the manuscript technically sound, and do the data support the conclusions?

Reviewer #1: Yes

Reviewer #2: Partly

2. Has the statistical analysis been performed appropriately and rigorously? 

Reviewer #1: Yes

Reviewer #2: Yes

3. Have the authors made all data underlying the findings in their manuscript fully available?

Reviewer #1: Yes

Reviewer #2: No

4. Is the manuscript presented in an intelligible fashion and written in standard English?

Reviewer #1: Yes

Reviewer #2: Yes

5. Review Comments to the Author

Reviewer #1: This is a well written article comparing keratoconus with high myopic astigmatism based on tomography data. This is of relevance in daily practice especially in the refractive sphere. Whilst well written, I query the originality of the paper and if it adds anything new to our understanding. I would recommend a further literature search to show what is known so far to give the reader context. Furthermore, whilst the data is comprehensive, it is presented in a laboroius manner and I would consider revising the manuscript by providing a summary.

Reviewer #2: dear Author most of my inquiries are from Materials and methods

1.“Patients were examined with a Scheimpflug principle-based Pentacam corneal topographer” suold be tomographer.

2.The Pentacam posses some very distinct parameters that are very specific for keratoconus and where not included in the paper: Anterior Radius of Curvature (ARC) Posterior Radius of Curvature (PRC) and on the BAD display Df,Db,Dp,Dt,De, BAD-D which were validated by Beilin and Ambrosio’s work and makes them the “go to” parameters to distinguish normal from keratoconus cornea when using the Pentacam

3.Way the particular parameters where chosen?

4.How was high myopic astigmatic patient were defined in diopters? And was the astigmatism matched to KC patients?

5.No details on inclusion and exclusion criteria like corneal interventions, use of contact lens and proper contact lens discontinuation before Pentacam scans etc.

6.Was there any randomization in choosing the patients in both groups ?

6. PLOS authors have the option to publish the peer review history of their article (what does this mean?). If published, this will include your full peer review and any attached files.

Reviewer #1: No

Reviewer #2: No

---

## [Author Response · Author response to Decision Letter 0]

28 Aug 2021

COMMENTS OF ACADEMIC EDITOR

COMMENT

“Please ensure that your manuscript meets PLOS ONE's style requirements, including those for file naming.” 

RESPONSE

Thank you. The suggested corrections are valid. The manuscript has been edited in the reference section to meet PLOS ONE’s style requirements and files have been named as requested. See red highlights on lines 287-373 of pages 16-20.

COMMENT

“Please provide additional details regarding participant consent. In the ethics statement in the Methods and online submission information, please ensure that you have specified what type you obtained (for instance, written or verbal, and if verbal, how it was documented and witnessed). If your study included minors, state whether you obtained consent from parents or guardians. If the need for consent was waived by the ethics committee, please include this information.” 

RESPONSE

We are very grateful for the academic editor’s constructive comment. We are sorry for the unfortunate mix up and the oversight of stating that the study employed a prospective design. The study rather used a retrospective design and covered a 7-year period from January 2014 to December, 2020. We are actually working on multiple articles that are based on similar datasets collected from the same facilities. Thus, the articles have got practically the same methods except that some are retrospective whereas others are prospective. We are sorry to say that there was a mix up of the methods of the current study protocol with that of another, and we assume responsibility for that and extend our apology. However, it is readily evident and can be confirmed from the discussion of our first submission that the current study is really retrospective as we stated the retrospective and non-randomized design of the study as a limitation. This statement has been highlighted in blue for ease of reference. Since medical records were reviewed retrospectively and identifying particulars of patients’ details concealed, patients’ consents were not needed for the study and were therefore waived by the IRB. Thank you so much for critical review which brought to bear this mix up.

See red highlights on lines 92-99 of page 5 and on lines 271-272 of page 15.

COMMENT

“In the Methods section of the manuscript please additional information regarding how the participants were recruited for the study included any eligibility criteria applied.” 

RESPONSE

Thank you. Since the study was retrospective and involved reviewing of medical records of already existing cases, there was no recruitment of subjects. See highlights on lines 92-99 of page 5. 

COMMENT

“You indicated that you had ethical approval for your study. In your Methods section, please ensure you have also stated whether you obtained consent from parents or guardians of the minors included in the study or whether the research ethics committee or IRB specifically waived the need for their consent.” 

RESPONSE

Thank you. Since medical records were reviewed retrospectively and identifying particulars of patients’ details concealed, patients’ consents were not needed for the study and were therefore waived by the IRB.

See red highlight on page 5, lines 92-99.

COMMENT OF REVIEWER 1

“This is a well written article comparing keratoconus with high myopic astigmatism based on tomography data. This is of relevance in daily practice especially in the refractive sphere. Whilst well written, I query the originality of the paper and if it adds anything new to our understanding. I would recommend a further literature search to show what is known so far to give the reader context. Furthermore, whilst the data is comprehensive, it is presented in a laboroius manner and I would consider revising the manuscript by providing a summary.”

RESPONSE

 Thank you. A further literature search has been conducted to show what is known so far to provide the reader with context. Also, even though the data appear too elaborate, it was presented in a manner that would enable easy comparison with similar studies, many of which presented their data in an equal fashion. (reference numbers 15, 18, 20, 22, 32). Besides, the tables have been split to form five tables with independent table headings. See highlighted changes on pages 3-4. Lines 53-78. Also see red highlights on lines 321, 329, 335, 340 and 368 of pages 18 and 20.

COMMENT OF REVIEWER 2

dear Author most of my inquiries are from Materials and methods 

“Patients were examined with a Scheimpflug principle-based Pentacam corneal topographer” suold be tomographer. 

RESPONSE

Thank you. “topographer” has been replaced with “tomographer” in the Materials and methods section. Kindly see highlighted changes on lines 109-110 of page 5.

COMMENT

‘The Pentacam posses some very distinct parameters that are very specific for keratoconus and where not included in the paper: Anterior Radius of Curvature (ARC) Posterior Radius of Curvature (PRC) and on the BAD display Df,Db,Dp,Dt,De, BAD-D which were validated by Beilin and Ambrosio’s work and makes them the “go to” parameters to distinguish normal from keratoconus cornea when using the Pentacam’ 

RESPONSE

Thank you. All the parameters you have drawn our attention to were actually included in the results section and written about in the discussion section of the main manuscript as well as in the abstract. 

Kindly see red highlights on tables 1, 2, 4 and 5 of pages 8, 9, 10, 11 and 12. Also see red highlights on lines 264-271 of page 15.

COMMENT 

“Way the particular parameters where chosen?” 

RESPONSE

Thank you very much.

All chosen parameters were informed by literature and related studies.

COMMENT

“How was high myopic astigmatic patient were defined in diopters? And was the astigmatism matched to KC patients?” 

RESPONSE

Thank you. This is a valid question. Patients who had sphero-cylindrical refractive errors with spherical components greater than -6.00D were considered high myopic astigmats. Besides, because keratoconus induces high and irregular astigmatism, it was difficult getting high myopic astigmats with matched cylinder components. However, cases in both groups were age matched.

See highlighted changes on lines 135-137 of page 7.

COMMENT

“No details on inclusion and exclusion criteria like corneal interventions, use of contact lens and proper contact lens discontinuation before Pentacam scans etc.” 

RESPONSE

Thank you. Inclusion and exclusion criteria have been expanded to include details on contact lens wear and corneal interventions prior to Pentacam scans. See highlighted changes on lines 104-108 of page 5.

COMMENT

“Was there any randomization in choosing the patients in both groups?” 

RESPONSE

Thank you. The study employed a retrospective cross-sectional design and did not require random assignment of participants. See red highlights on lines 92-99 of page 5.

---

## [Decision Letter · Decision Letter 1]

15 Nov 2021

The utility of measures of anterior segment parameters of pentacam scheimpflug tomographer in discriminating high myopic astigmatism from keratoconus

PONE-D-21-20839R1

Dear Dr. Kyei,

We’re pleased to inform you that your manuscript has been judged scientifically suitable for publication and will be formally accepted for publication once it meets all outstanding technical requirements.

Kind regards,

Michael Mimouni

Academic Editor

PLOS ONE

Additional Editor Comments (optional):

Reviewers' comments:

Reviewer's Responses to Questions

**Comments to the Author**

1. If the authors have adequately addressed your comments raised in a previous round of review and you feel that this manuscript is now acceptable for publication, you may indicate that here to bypass the “Comments to the Author” section, enter your conflict of interest statement in the “Confidential to Editor” section, and submit your "Accept" recommendation.

Reviewer #1: All comments have been addressed

Reviewer #2: All comments have been addressed

2. Is the manuscript technically sound, and do the data support the conclusions?

Reviewer #1: Yes

Reviewer #2: Yes

3. Has the statistical analysis been performed appropriately and rigorously? 

Reviewer #1: Yes

Reviewer #2: Yes

4. Have the authors made all data underlying the findings in their manuscript fully available?

Reviewer #1: Yes

Reviewer #2: Yes

5. Is the manuscript presented in an intelligible fashion and written in standard English?

Reviewer #1: Yes

Reviewer #2: Yes

6. Review Comments to the Author

Reviewer #1: The authors have adequately responded to the issues raised in the previous review. This is a relevant paper which the readership will find useful.

Reviewer #2: Dear Author, I appreciate your dedicated and thorough response.

you addressed all of my questions and modified the the paper accordingly.

7. PLOS authors have the option to publish the peer review history of their article (what does this mean?). If published, this will include your full peer review and any attached files.

Reviewer #1: No

Reviewer #2: No

---

## [Editor Report · Acceptance letter]

17 Nov 2021

PONE-D-21-20839R1 

The utility of measures of anterior segment parameters of a Pentacam Scheimpflug tomographer in discriminating high myopic astigmatism from keratoconus 

Dear Dr. Kyei:

I'm pleased to inform you that your manuscript has been deemed suitable for publication in PLOS ONE. Congratulations! Your manuscript is now with our production department. 

Kind regards, 

on behalf of

Dr. Michael Mimouni 

Academic Editor

PLOS ONE